# RECURSIVE BINARY NEURAL NETWORK LEARNING MODEL WITH 2-BIT/WEIGHT STORAGE REQUIREMENT

## ABSTRACT

This paper presents a storage-efficient learning model titled Recursive Binary Neural Networks for embedded and mobile devices having a limited amount of on-chip data storage such as hundreds of kilo-Bytes. The main idea of the proposed model is to recursively recycle data storage of weights (parameters) during training. This enables a device with a given storage constraint to train and instantiate a neural network classifier with a larger number of weights on a chip, achieving better classification accuracy. Such efficient use of on-chip storage reduces off-chip storage accesses, improving energy-efficiency and speed of training. We verified the proposed training model with deep and convolutional neural network classifiers on the MNIST and voice activity detection benchmarks. For the deep neural network, our model achieves data storage requirement of as low as 2 bits/weight, whereas the conventional binary neural network learning models require data storage of 8 to 32 bits/weight. With the same amount of data storage, our model can train a bigger network having more weights, achieving 1% less test error than the conventional binary neural network learning model. To achieve the similar classification error, the conventional binary neural network model requires 4× more data storage for weights than our proposed model. For the convolution neural network classifier, the proposed model achieves 2.4% less test error for the same on-chip storage or 6× storage savings to achieve the similar accuracy.

## 1 INTRODUCTION

Deep Neural Networks (DNN) have demonstrated the state-of-the-art results in a wide range of cognitive workloads such as computer vision Krizhevsky et al. (2012) and speech recognition (Hinton et al. (2012)), achieving better-than-human performance for the tasks often considered too complex for machines. The success of DNNs has indeed motivated scientists and engineers to implement a DNN in mobile and embedded devices, dubbed as Internet of *Smart* Things (Kortuem et al. (2010)). The recent works in this area, however, mostly implement the inference function of DNN, rather than training, while training is performed in cloud computers and post-training weights are downloaded to mobile and embedded devices (Lane et al. (2016)).

*On-device* learning, however, becomes increasingly important for the mobile and embedded devices for the following three reasons. First, an intelligent device benefits to have the model that is custom-built for the device itself, its end user, and environment. This is because the model tends to be more accurate and effective if constructed with the consideration of those factors. Second, the training data from mobile and embedded devices can contain security-sensitive information, e.g., personal health data from wearable medical devices. At the risk of being leaked, users typically do not want to upload such data onto cloud computers. Finally, in the era of Internet of Things (IoT), we anticipate a drastic increase in the number of deployed devices, which can proportionally increase the number of learning tasks to be done in the cloud. Coupled with the complexity of training, even for powerful cloud computers, this can be a computationally challenging task.

On-device learning, however, entails various challenges in algorithms, data, and systems (Roschelle (2003); Vogel et al. (2009)). The most eminent challenge regarding computing systems is high energy consumption caused by dense computation and data access, which is considered prohibitive for the limited resources of embedded devices. The high overhead of data access is caused by

fetching DNN weights from DRAM (or FLASH) external to a computing chip on an embedded device. Since the data storage size is limited for such computing chip, the parameters of a DNN have to be stored in external DRAM and FLASH during training. For example, ARM Cortex M3 processor, a processor widely used in commercial wearable devices such as FitBit, has only 64 kilo-Byte (kB) on-chip data storage. This can only store very small size of DNN especially if each weight is 32-bit float point number. Compared to accessing on-chip SRAM, accessing off-chip DRAM incurs 3 to 4 orders of magnitudes more energy and delay overhead. Therefore, fetching weights every time for each data makes training prohibitive to be implemented on a mobile and embedded device (Han et al. (2015)).

Recently several techniques such as pruning, distilling, and binarizing weights have been proposed to compress the parameters of a DNN. This makes it more feasible to fit weights in on-chip SRAM (Han et al. (2015); Courbariaux et al. (2015; 2016); Rastegari et al. (2016); Hinton et al. (2015)). These techniques can also reduce computation overhead. However, these works focused on weight size compression ***after training is finished***. The data storage requirement ***during training*** remains the same.

Similarly, several learning models, which belong to so-called Binary Neural Networks (BNN), have been proposed (Courbariaux et al. (2015; 2016); Rastegari et al. (2016)). These model uses sign bits (or binary information) of weights in several parts of the learning model notably the part of multiplying and accumulating weights with inputs/activations. Although this greatly reduces computational complexity, each weight still needs to be represented in high precision number with multiple bits (e.g. 32 bits in Courbariaux et al. (2015; 2016); Rastegari et al. (2016)) during the end-to-end training process. This is because weights have to be fine-tuned in the weight update part. Therefore, this so-called BNN models have not demonstrated to scale storage requirement for training below 32 bits/weight.

Our goal is, therefore, to efficiently use the limited amount of on-chip data storage during training. We also aim to scale computational complexity. Toward this goal, we propose a new learning model, *Recursive Binary Neural Network (RBNN)*. This model is based on the process of weight training, weight binarization, recycling storage of the non-sign-bit portion of weights to add more weights to enlarge the neural network for accuracy improvement. We recursively perform this process until either accuracy stops improving or we use up all the storage on a chip.

We verified the proposed RBNN model on a Multi-Layer Perceptron (MLP)-like and a convolutional neural network classifier on the MNIST and Voice Activity Detection (VAD) benchmark. We considered typical storage constraints of embedded sensing devices in the order of hundreds of kB. The experiment in the MLP-like classifier on MNIST confirms that the proposed model (i) demonstrates 1% less test error over the conventional BNN learning model specifically following Courbariaux et al. (2015) for the same storage constraints or (ii) scales on-chip data storage requirement by $4\times$ for the same classification test error rate($\sim$2%), marking the storage requirement of 2 bits/weight. The conventional BNN models in Courbariaux et al. (2015; 2016); Rastegari et al. (2016) exhibit a significantly larger storage requirements of 8 to 32 bits/weight. The experiment of the CNN classifier for MNIST confirms up to $6\times$ reduction of data storage requirement and 2.4% less test error. For the VAD benchmark, the proposed RBNN achieves $9\times$ savings in data storage requirement.

The remainder of the paper is as follow. In Sec. 2 we will introduce the works related to this paper, including comparison to existing works on distillation, compression, BNNs, and low-precision weights. In Sec. 3 we will describe our proposed model. Sec. 4 will present the experimental results and comparisons to the conventional BNN model. Finally, in Sec. 5, we will conclude the paper. The paper includes Appendix A to D to describe additional experiments and analysis.

## 2 RELATED WORK

### 2.1 DISTILLATION AND COMPRESSION OF DNN PARAMETERS

Knowledge distillation (Hinton et al. (2015)) is a technique to compress knowledge of an ensemble of DNNs into one small DNN while maintaining the accuracy. Although this technique can scale the number of weights for deployment systems post-training, it cannot scale data storage requirement

for training. Specifically, during training, each of weights is represented in high-precision number, which needs to be stored in multi-bit data storage.

Another technique is to compress the data size of weights by exploiting redundancies in them. In Han et al. (2015), the authors combine four sub-techniques, namely weight pruning, quantization, sharing, and compression coding to reduce the data size of weights. Similar to the knowledge distillation, this technique can be applied to the weights that are already trained, and cannot scale data storage requirement of weights during training.

## 2.2 BINARY NEURAL NETWORK (BNN)

Recent works proposed to use binary information of weights (Courbariaux et al. (2015); Baldassi et al.), activations (Courbariaux et al. (2016); Rastegari et al. (2016)), and even inputs (Rastegari et al. (2016)) in some parts of training and post-training operations. The use of binary information of weights notably in Multiply-and-Accumulate (MAC) operation can drastically reduce computational complexity. However, those BNN techniques still cannot scale the storage requirement of weights during training. In these works, each weight is represented in 32 bits. This is because mainstream training models such as stochastic gradient decent requires to update weights in a fine-grained manner.

## 2.3 LOW-PRECISION FIX-POINT WEIGHT REPRESENTATION

Several studies have demonstrated that moderately lowering the precision of weights (i.e., quantization) has a tolerable impact on training and post-training operations of DNN (Gupta et al. (2015); Courbariaux et al. (2014)). In Gupta et al. (2015), the authors trained a DNN having 16-bit fixed-point weights with the proposed stochastic rounding technique, and demonstrated little to no degradation in classification accuracy. In Courbariaux et al. (2014), the authors proposed the dynamic fixed-point representation (i.e., dynamically changing the position of decimal point over computation sequences) to further reduce the precision requirement down to 10 bits per synapse. Using fixed-point representation help to reduce storage requirement and fixed-point arithmetic is more hardware friendly (Han et al. (2015)).

# 3 RECURSIVE BINARY NEURAL NETWORK (RBNN) MODEL

## 3.1 KEY IDEA

Table 1 shows which information of weights are used in each step of training in both conventional BNN Courbariaux et al. (2015; 2016); Rastegari et al. (2016) and our proposed RBNN. The conventional BNN works (Courbariaux et al. (2015; 2016); Rastegari et al. (2016)) use sign bits of weights during multiply-and-accumulate (MAC) operation in forward and backward propagations. However, the weight update has to be done with high precision. This mandates to store multi-bit (16 or 32 bits in those works) weights in data storage during learning, resulting in no savings in weight storage requirement. On the other hand, it has been studied that in the trained neural networks we can use only the sign bits of weights to perform inference (Courbariaux et al. (2015; 2016); Rastegari et al. (2016)). This vast difference in the requirements of weight precision between learning and post-learning inspires us to create our RBNN model.

As shown in Table 1, we also use only the sign bits for MAC operations to reduce computational complexity for training. The main difference is that we binarize weights (keep only the sign bits) and then we recycle the data storage that are used to store these non-sign bits of weights. This recycled storage is used to add more multi-bit *trainable* weights to the neural network. We then train this new network having both the binarized non-trainable weights and the newly-added trainable weights. We perform these steps recursively, which makes the neural networks larger and more accurate but using the same amount of data storage for weights.

Figure 1 depicts the process of our proposed RBNN learning model with an example of the multi-layer neural network classifier. In the beginning, the neural network has one input, two sets of two hidden, and one output neurons, and eight weights each of which has n bits. We first train this $1 \times 2 \times 2 \times 1$ network using the conventional back-propagation training algorithm for BNN

Table 1: Comparisons of weight information usage in BNNs and RBNN

| Steps | BNN | Proposed RBNN |
|---|---|---|
| **MAC in forward prop.** | Sign bits of weights | Sign bits of weights |
| **MAC in back prop.** | Sign bits of weights | Sign bits of weights |
| **Weight update** | All bits of weights | All bits of weights |
| **Recursive recycling** | N/A | Keep sign bits and recycle storages of the other bits for more plastic weights |

(Courbariaux et al. (2015)). After that, we discard all bits except the sign bit in each weight (binarization), resulting in a $1 \times 2 \times 2 \times 1$ trained network having binary weights (*trained_BNN*). Then we continue the second iteration of training (the second subfigure of Figure 1). Specifically, we recycle the storage that is used to store the n-1 non-sign bits of weights in the $1 \times 2 \times 2 \times 1$ network. Using this data storage, we add a new network named *incremental_BNN* comprising eight additional weights ($W_{21}$ to $W_{28}$ in Figure 1) to the *trained_BNN*, expanding the network size to $1 \times 4 \times 4 \times 1$ which we name as *enlarged_BNN*. In the *enlarged_BNN*, each of the newly-added weights is $n-1$ bits. In other words, the *enlarged_BNN* comprises of one *trained_BNN* that has eight weight ($W_{11}^b$ to $W_{18}^b$) that are trained (binary, non-plastic, marked as solid lines in Figure 1) and one *incremental_BNN* with eight weights ($W_{21}$ to $W_{28}$) that are under training (n-1 bits, plastic, marked as dash lines in Figure 1). The *incremental_BNN* is trained together with the *trained_BNN* but only the weights of *incremental_BNN* are updated.

We repeat the same process of binarization and recycling. In every iteration, the *enlarged_BNN* integrates eight more weights, and the bit-width of newly-added plastic weights in the *incremental_BNN* is reduced by one. At the k-th iteration, the *trained_BNN* has $8 \cdot (k-1)$ neurons and the plastic weights have $(n-k+1)$ bit-width. After the k-th iteration, as shown in the rightmost in Figure 1, the neural network becomes a $1 \times 2k \times 2k \times 1$ with $8 \cdot k$ binary weights. This network has k times more weights than the first $1 \times 2 \times 2 \times 1$ network. However, the data storage used for weights remains the same, scaling the storage requirement per weight to $n/k (= 4 \cdot n/4 \cdot k)$, which is k times smaller than that of the first network. Thus the proposed RBNN can either achieve better classification accuracy - enabled by the more number of weights - with the same amount of weight storage, or reduce weight storage requirement for the same classification accuracy level.

## 3.2 MODEL DETAILS

Figure 2 depicts the details of the proposed RBNN model. In the beginning of the training procedure, conventional BNN training algorithm *BNN_Training* is used to train a BNN. After training, we have a *trained_BNN* having binary weights. Then we reduce the weight bit-width by one and train a new *incremental_BNN*. The training algorithm for *incremental_BNN* is named as *incremental_BNN_Training* which is shown in Algorithm 1. After the *incremental_BNN* is trained, the performance of the *enlarged_BNN* is tested. If the accuracy keeps on improving and there is still available data storage after weight binarization, we will continue to reduce the weight bit-width by one and train another *incremental_BNN*.

The method *Incremental_BNN_Training* is designed to train the *incremental_BNN* to improve performance of *enlarged_BNN*. It is based on the conventional BNN training method. As shown in Algorithm 1, the main idea of this training method is: both *trained_BNN* and *incremental_BNN* are used to calculate the output of the *enlarged_network* in the forward propagation. During back-propagation and parameter-update, however, only plastic weights in *incremental_BNN* are updated. The binary weights in *trained_BNN* are not modified. One possible hardware and software implementation of this sub-word operation of synaptic weights are illustrated in Appendix A. Note that similar to the conventional BNN training algorithm (Zhou et al. (2002)), binary weights are used in both forward and backward propagation in *Incremental_BNN_Training*, to reduce computational overhead. Since weights in *trained_BNN* are binary, the multiplication related to weights are simplified as shift.

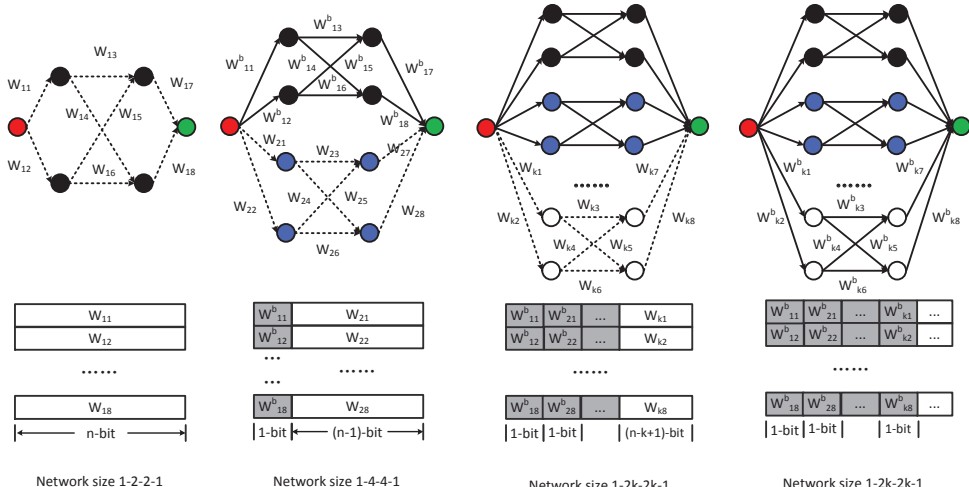

Figure 1: RBNN learning model with an example neural network. The recursive operation increases the number of weights in the neural network (top) while using the same amount of storage for weights (bottom).

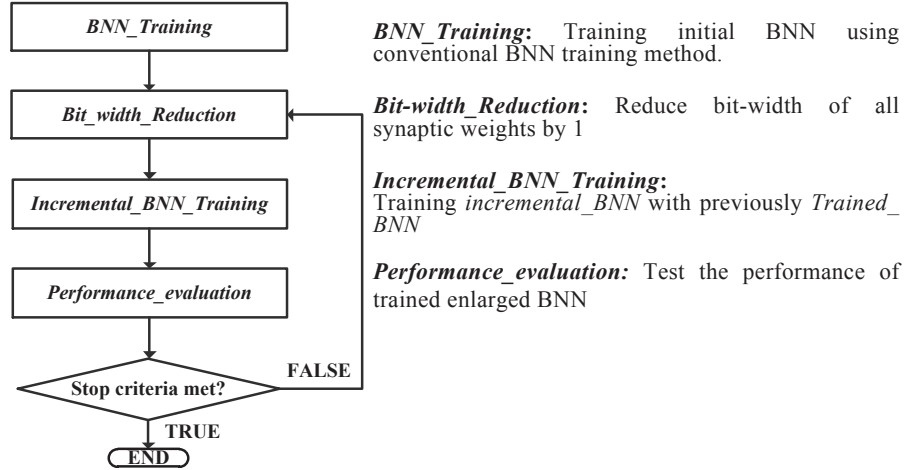

Figure 2: RBNN training flowchart.

**BNN_Training**: Training initial BNN using conventional BNN training method.

**Bit-width_Reduction**: Reduce bit-width of all synaptic weights by 1

**Incremental_BNN_Training**: Training *incremental_BNN* with previously *Trained_BNN*

**Performance_evaluation**: Test the performance of trained enlarged BNN

## 4 EXPERIMENT SETUP

In this and the next section, we will describe the detailed experiment setup and the results for the MLP-like classifier and the MNIST benchmark. In addition, we will discuss the setup and results of applying the proposed RBNN model to CNN classifiers and VAD benchmarks in Appendix B and C, respectively.

### 4.1 PERMUTATION-INVARIANT MNIST BENCHMARK

We used the permutation-invariant MNIST to test the performance of the proposed RBNN model on MLP-like classifier. We use the original training set of 60,000 28-by-28 pixel gray-scale images and the original test set of 10,000 images. The training and testing data are normalized to [-1, 1] and zero mean. Following the common practices, we use the last 10,000 images of the training set as a validation set for early stopping and model selection. We did not consider data augmentation, pre-processing, and unsupervised pre-training during our experiment.

---

**Algorithm 1** *Incremental_BNN_Training*. $C$ is the cost function for mini-batch, $\eta$ the learning rate and L the number of layers. The function *Binarize()* specifies how to binarize the weights. *Act_hid()* and *Act_out()* are activation function of hidden layers and output layer, respectively.

---

**Require:** a minibatch of inputs and targets $(a_0, a^*)$, previous weights of incremental BNN $W(I)$, weights of *trained_BNN* $W(T)$

**Ensure:** updated weights of incremental BNN $W(I)^{(t+1)}$

  **1. Forward Propagation**

  1.1 Computing outputs of hidden layers in *trained_BNN* and *incremental_BNN*

  **for** k = 1 to L-1 **do**

      $a(T)_k = Act\_hid(W(T)_k \cdot a(T)_{(k-1)})$

      $W(I)_k^b \leftarrow Binarize(W(I)_k^b)$

      $a(I)_k = Act\_hid(W(I)_k^b \cdot a(I)_{(k-1)})$

  **end for**

  1.2 Computing outputs of enlarged BNN

  $a_L = Act\_out(W(T)_L \cdot a(T)_{(L-1)} + W(I)_L \cdot a(I)_{(L-1)})$

  **2. Backward propagation**

  {Please note that only gradients of *incremental_BNN* are computed.}

  Compute $g_{aL} = \frac{\partial C}{\partial a_L}$ knowing $a_L$ and $a^*$

  **for** k = L to 1 **do**

      $g_{W(I)_k^b} \leftarrow (g_{a(I)_k} \circ a'(I)_k) \cdot (W(I)_k^b) \cdot a(I)_{k-1}$

  **end for**

  **3. Parameter Update**

  Please note that only weights of incremental_BNN are updated.

  **for** k = L to 1 **do**

      $W(I)_k^{T+1} \leftarrow W(I)_k^t + \eta \cdot g_{WI_k^b}$

  **end for**

---

## 4.2 NEURAL NETWORK CONFIGURATION AND DATA FORMAT

We consider the storage constraints of mainly hundreds of kB based on the typical embedded systems (Shiue & Chakrabarti (1999)). We considered a feed-forward neural network with one or two hidden layers. We considered several different numbers of neurons in the hidden layer ranging from 200 to 800. The numbers of the input and output units are 784 and 10, respectively. We used the *tanh_opt()* for the activation function of the hidden layer and the *softmax()* or linear output for that of the output layer. We used the classical Stochastic Gradient Descent (SGD) algorithm for cross-entropy or hinge loss minimization without momentum. We used a small size of batch (1,000) and a single static learning rate which is optimized for each BNN. Any other advanced techniques such as dropout, Maxout, and ADAM are not used for both the proposed and the baseline learning models. We recorded the best training and test errors associated with the best validation error after up to 1,000 epochs. The results from 20 independent experiments are averaged for each case.

We used the fixed-point arithmetic for all the computation and data access. The fixed-point intermediate computations, such as gradient calculation, also use fixed-point arithmetic with sufficient precision. The translations from wide fixed-point numbers to narrow fixed-point and binary numbers are performed with simple decimation without using advanced techniques such as stochastic rounding (Courbariaux et al. (2014)). We saturated values in the event of overflow or underflow in weight update. The dynamic range of fixed-point representation is optimized to achieve better accuracy performance.

## 5 RESULTS AND DISCUSSION

### 5.1 ACCURACY IMPROVEMENT

Figure 3 depicts the classification errors of the proposed RBNN model across three recursive iterations. The initial bit-width of weights is eight. In each series of data points in Figure 3, the leftmost point represents the initial neural network, i.e., with 2 layers of 200 hidden units and

198,800 weights ($= 784 \cdot 200 + 200 \cdot 200 + 200 \cdot 10$). At this point, the storage requirement, defined as the ratio of total storage bits to the number of weights, is 8 bits/weight. The network at this point, is equivalent to one trained by the conventional BNN model specifically following Courbariaux et al. (2015). The second leftmost data point in the series is the neural network after the first recursive iteration. The network size is enlarged by twice, resulting in the $784 \times 400 \times 400 \times 10$ network. This reduces storage requirement to 4 bits/weight. Compared to the initial BNN, the *enlarged_BNN* achieves $\sim 0.7\%$ and $\sim 0.4\%$ reduction in training and test error rate, respectively. Finally, after three recursive iterations, the size of the neural network becomes $784 \times 800 \times 800 \times 10$ (555,800 weights). It marks the storage requirement as small as 2 bits/weight to achieve the test error of 2.17%. This accuracy is as good as the fully-connected network using $4 \times$ times more data storage for weights, trained by the conventional BNN model in (Courbariaux et al. (2015)).

Note that we have various ways of using the recycled data storage to enlarge the neural network in the proposed RBNN model. As shown in Figure 1, we chose a "tiled" approach where no connections are made among *incremental_BNN*s. This is because it is easier to implement the algorithm in hardware or map it onto the conventional CPU and GPU (see Appendix A). In Appendix D, we have the RBNN to train a fully-connected DNN. The results show that with same size of total data storage for weights, both *tiled* and fully-connected exhibit similar test error.

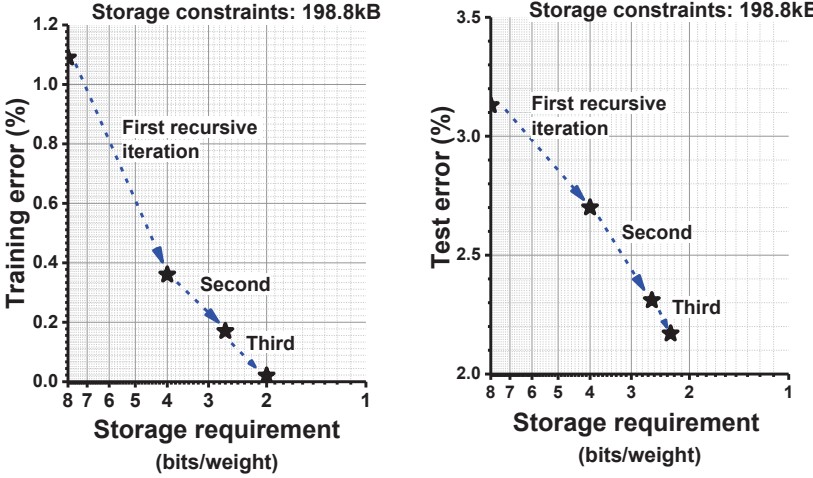

Figure 3: (left) Training error and (right) test error across recursive iterations in the proposed RBNN model. The total weight storage assumed in this experiment is 198.8 kB.

## 5.2 STORAGE AND ARITHMETIC COMPLEXITY

To evaluate the storage and arithmetic complexity of the proposed RBNN, we trained multiple single-hidden-layer DNNs using the proposed RBNN and the conventional BNN model (Courbariaux et al. (2015)). For the conventional model, we considered BNN containing 100 to 800 hidden neurons and 6 to 16 bit weight precisions. For the proposed model, we considered 100 to 800 initial hidden neurons and 12 to 16 bit initial weight precisions. Those DNNs require 116 kB to 1.2 MB data storage for weights.

Figure 4 shows the results of this experiment: the proposed model can achieve 1% less test error than the conventional model using the similar amount of data storage. To achieve the similar test error, the proposed RBNN model requires 3-4$\times$ less data storage than the conventional BNN model.

Table 2 shows the detail comparisons of six neural networks out of the 16 networks shown in Figure 4, three of which are trained by the proposed RBNN model ($R_1$, $R_2$, $R_3$) and the other three by the conventional BNN model ($B_1$, $B_2$, $B_3$) (Courbariaux et al. (2015)). We compare the arithmetic complexity for training and inferring. For training, to achieve similar accuracy performance ($R_1$

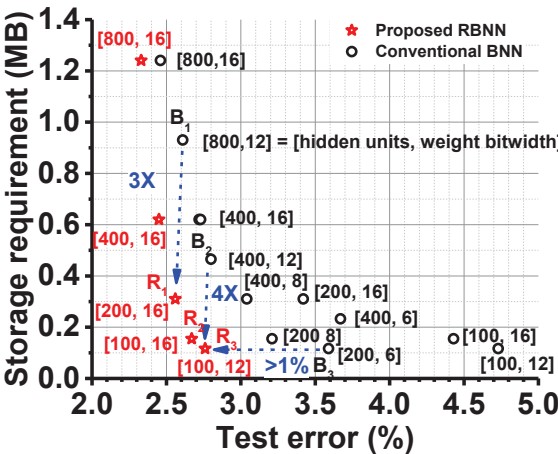

Figure 4: The storage requirement and test error trade-offs achieved by the proposed RBNN model and the conventional BNN model. The proposed model achieves $3\times$ data storage savings for the same test error and $> 1\%$ lower error for the same data storage.

Table 2: Detail comparisons of RBNNs and BNNs

|  | $R_1$ | $R_2$ | $R_3$ | $B_1$ | $B_2$ | $B_3$ |
|---|---|---|---|---|---|---|
| Initial hidden neurons | 200 | 100 | 100 | 800 | 400 | 200 |
| Final hidden neurons | 800 | 700 | 400 | 800 | 400 | 200 |
| Final synaptic weights | 635,200 | 555,800 | 317,600 | 635,200 | 317,600 | 155,600 |
| Initial weight bit-width | 16 | 16 | 12 | 12 | 12 | 16 |
| Storage requirement | **4** | 2.28 | **3** | **12** | **12** | 16 |
| Test error (%) | 2.56 | 2.65 | 2.76 | 2.61 | 2.80 | 3.60 |
| Arithm., training | 2,223,200 | 2,779,000 | 1,111,600 | 1,270,400 | 635,200 | 317,600 |
| Shift/Multiply/Add | **635,200** | 555,800 | **317,600** | **635,200** | **317,600** | 158,800 |
|  | 2,223,200 | 2,779,000 | 1,111,600 | 1,270,400 | 635,200 | 317,600 |
| Arithm., inference | **635,200** | 555,800 | **317,600** | **635,200** | **317,600** | 158,800 |
| Shift,Add | **635,200** | 555,800 | **317,600** | **635,200** | **317,600** | 158,800 |
| Storage for weights | 310kB | 155kB | 116kB | 930kB | 465kB | 114kB |
| Total Train Energy (nJ) | **2,715.18** | 2,459.41 | **1,004.58** | **231,197.91** | **115,304.86** | 655.05 |
| Arith. | **365.24** | 402.95 | **123.56** | **175.67** | **87.84** | 67.49 |
| Data Access | **2,350.24** | 2,056.46 | **881.02** | **231,022.24** | **115,217.02** | 587.56 |

Table 3: Energy table for 45nm CMOS process

| **Operation(int)** | 12-bit ADD/ SHIFT | 12-bit MULT | 12-bit SRAM | 12-bit DRAM | 16-bit ADD/ SHIFT | 16-bit MULT | 16-bit SRAM | 16-bit DRAM |
|---|---|---|---|---|---|---|---|---|
| Energy [pJ] | 0.0375 | 0.126 | 1.387 | 240 | 0.05 | 0.225 | 1.85 | 320 |
| Relative Cost | 1 | **3.4** | **37** | **6400** | 1.3 | **6** | **49.3** | **8533** |

and $B_1$; $R_3$ and $B_2$), RBNN requires around twice as many add and shift operations as conventional BNN does. On the other hand, RBNN and BNN have the same amount of multiply operations. Since the complexity of multiplication is much higher than add and shift, it is important not to increase the

number of multiplications. For inference, both RBNN and BNN have the same amount of shift and add operations. Inference requires no multiplication since MAC uses binary information of weights.

## 5.3 ENERGY CONSUMPTION SAVINGS

In Table 2, we also compare the energy dissipations for training. Total energy dissipation per one data and one epoch $E_{total}$ is:

$$E_{total} = E_{arith} + E_{acc} \tag{1}$$

, where $E_{arith}$ is the energy dissipation of arithmetic operations and $E_{acc}$ is the energy dissipation of data storage access for weights. $E_{arith}$ is:

$$E_{arith} = N_{shift} \cdot E_{shift} + N_{add} \cdot E_{add} + N_{mult} \cdot E_{mult} \tag{2}$$

, where $N_{shift}$, $N_{add}$, and $N_{multiply}$ are the numbers of shifts, adds, and multiplications, respectively, and $E_{shift}$, $E_{add}$, and $E_{mult}$ are their energy consumptions. $E_{acc}$ is calculated as:

$$E_{acc} = (2 \cdot N_{weight,SRAM} \cdot E_{access,SRAM} + 2 \cdot N_{weight,DRAM} \cdot E_{access,DRAM}) \cdot N_{iteration} \tag{3}$$

, where $N_{weight,SRAM}$ and $N_{weight,DRAM}$ are the number of SRAM and DRAM accesses, respectively, and $E_{access,SRAM}$ and $E_{access,DRAM}$ are their respective energy dissipations. $N_{iteration}$ is the number of recursive iterations in the RBNN and it becomes 1 in the conventional BNN training model. 2 is factored since weights are accessed two times in forward and backward propagations.

Table 3 summarizes energy cost of each operation. It is based on the 45nm CMOS process, presented in Han et al. (2016). We normalized the energy costs to the bit-widths of operations, quadratically for multiplication and linearly for all the other operations. DRAM access consumes $173\times$ more energy than SRAM access, and $1,422\times$ than multiplication. Therefore, it is critical to reduce DRAM access for saving energy. In the conventional BNN traning case, however, we have to store the extra weights that cannot be stored in SRAM in DRAM. Our RBNN, however, can utilize only SRAM for weight access during the training process. This difference results in $\sim100\times$ less energy dissipation in the RBNN.

## 6 CONCLUSION AND FUTURE WORK

This paper presents a new learning model for on-device training with limited data storage. The proposed RBNN model efficiently uses limited on-chip data storage resources by recycling the part of data storage that would have been wasted in conventional BNN model, to add and train more weights to a neural network classifier. We verified the proposed model with MLP-like DNN and CNN classifiers on the MNIST and VAD benchmark under the typical embedded device storage constraints. The results of MLP-like DNNs on MNIST show that the proposed model achieves 2 bits/weight storage requirement while achieving 1% less test error as compared to the conventional BNN model for the same storage constraint. Our proposed model also achieves $4\times$ less data storage than the conventional model for the same classification error. The similar to greater savings are verified with the CNN classifiers and the VAD benchmarks. We expect the future work of further reduce computation complexity, such as binarization of activation function of BNN (Courbariaux et al. (2016)). We also expect to apply the RBNN model to the ensembles of neural networks (Zhou et al. (2002), and the mixture of experts (Shazeer et al. (2017)).

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

## A    IMPLEMENTATION OF SUB-WORD OPERATION OF RBNN

In the proposed RBNN model, each word contains both non-plastic and plastic weights but we need to update only the plastic weights. We can implement this sub-word operation using the mask and bitwise logical operations which are widely supported in the conventional CPUs and GPUs. Figure 5 illustrates a possible implementation. We assume the word size of eight bits, where at the exemplary moment three already-trained weights take up the top three bits of the word and the weight currently under training takes the remaining five bits. We fetch this weight word from the storage. We also generate/fetch a mask word *synp_mask* which stores 11100000 in this example. We bitwise-AND the weight word with the mask word to produce a temporary word *synp_fix*. We do the same with the bitwise inverse of the mask word to another temporary word *synp_plsb*, which is then updated via the RBNN model. Note that *synp_fix* is not changed. These two words, then, combined via a bitwise XOR operation to produce a word *synp_out*. This completes one training epoch. As compared to the conventional BNN model, the proposed RBNN additionally requires only two bitwise-AND and one bitwise-XOR operations. These operations are supported in modern CPUs and GPUs and their cost is not significant.

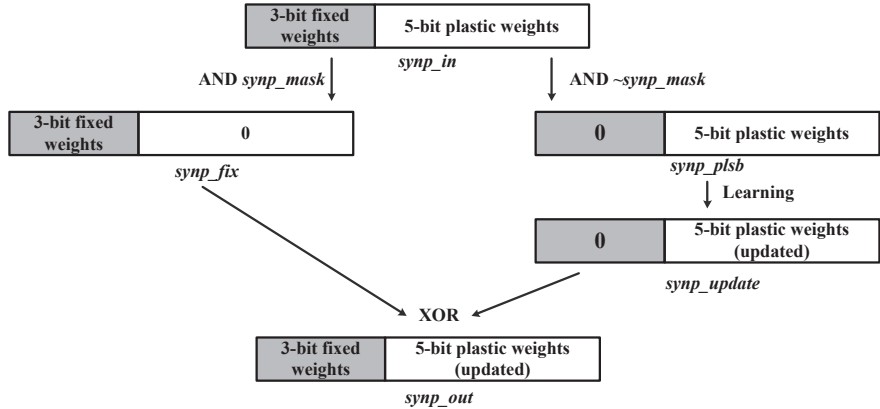

Figure 5: Sub-word operation using a mask word and bitwise AND and XOR operations in RBNN.

## B    APPLICATION OF RBNN TO CONVOLUTIONAL NEURAL NETWORKS

### B.1    EXPERIMENT SETUP

We applied the proposed RBNN model to the LeNet5 Convolutional Neural Network (LeCun et al. (1998)). The network has two convolution layers, one having six 5-by-5 and the other having twelve 5-by-5 feature maps. Each of the convolution layers is followed by a $4\times$ dowsampling average-pooling layer. The LeNet5 has a fully-connected (FC) classifier consist of one input, one hidden, and one output layer. As in Courbariaux et al. (2015), we used binary information of weights in the convolutional layers and the FC classifier for forward and backward propagations and fixed-point weights for weight update. We applied the proposed RBNN model on the hidden layer of the FC classifier.

### B.2    RESULTS

We trained multiple CNN classifiers for the MNIST benchmark while changing configurations of the FC classifier. For the proposed RBNN model, we considered the FC classifier containing 200 to 800 *initial* hidden neurons and 16 bit *initial* weight precision. For the conventional BNN model Courbariaux et al. (2015), we considered the FC classifier containing 200 to 2,500 hidden neurons and 16-bit weight precision. Those CNNs require 81 kB to 1.01 MB data storage for all the weights in the convolutional layers and the FC classifer. Figure 6 shows the trade-off between the test error and the weight storage requirement of those CNNs. The proposed RBNN model can achieve 2.4% less test error than the conventional BNN model for the same amount of data storage for weights.

For the similar test error of ∼2.5%, the proposed RBNN model requires more than 6× less weight data storage than the conventional BNN model.

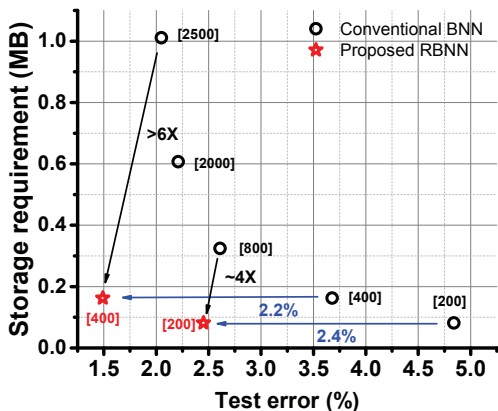

Figure 6: Trade-offs between test error and weight storage requirement. The proposed RBNN achieves more than 6X data storage savings for the same test error and ∼2.4% less test error for the same data storage requirement.

## C  APPLICATION OF RBNN TO VOICE ACTIVITY DETECTION

We applied the RBNN to train MLP-like DNN classifiers for the VAD benchmark. The VAD benchmark is based on Aurora 4 (Pearce & Picone (2002)), which has 7,133 utterances from 83 speakers. It also contains five noise scenarios: bus, cafe, park, river, and traffics. The signal-to-noise ratio of the data used in the experiment is 10 dB. We use the same DNN configurations used in Sec. 4.2. The input to the DNN (features) are five frames of 16-dimensional band-pass filter-bank output commonly used in other works Zhang & Wang. Table 4 summarizes the classifier models trained by the RBNN and the conventional BNN methods. For each noise scenario we list only the models that achieve the similar test errors. The experiment confirms that the proposed RBNN model can save up to 9× data storage than the conventional BNN for the similar level of detection accuracy.

Table 4: Accuracy and data storage size comparison of the RBNN and the conventional BNN on VAD benchmark

| Scenario | RBNN | | | BNN | | | Data storage savings |
| --- | --- | --- | --- | --- | --- | --- | --- |
| | Weight bit-width | Hidden neurons inital/final | Test accuracy(%) | Weight bit-width | Hidden neurons | Test accuracy(%) | |
| **bus** | | 100/500 | 5.27 | | 900 | 5.9 | **6.7×** |
| **cafe** | | 100/400 | 8.8 | | 1100 | 8.71 | **8.25×** |
| **park** | 16 | 100/600 | 7.94 | 12 | 1200 | 8.21 | **9×** |
| **river** | | 100/500 | 8.15 | | 1000 | 8.12 | **7.5×** |
| **traffic** | | 100/600 | 8.05 | | 900 | 8.07 | **6.75×** |

## D  RBNN IN FULLY-CONNECTED DNN SYSTEMS

In Sec. 3.1, we have the RBNN to train a *tiled* feedforward DNN. In this section, we experiment to train a *fully-connected* DNN using the proposed RBNN. Note that the fully-connected DNN is only one way of many other possible approaches on how to recycle the data storage to expand a neural network. Figure 7 illustrates the training process. It starts with an exemplary DNN whose initial

size is $1\times2\times2\times1$. Each weight is n bits. As shown in the first two sub-figures, we first train a tiled DNN as we did in Sec. 3.1. Then, we start to add the weights that connect between tiles (the last two sub-figures), again by recycling the data storage from the binarization in each recursive iteration.

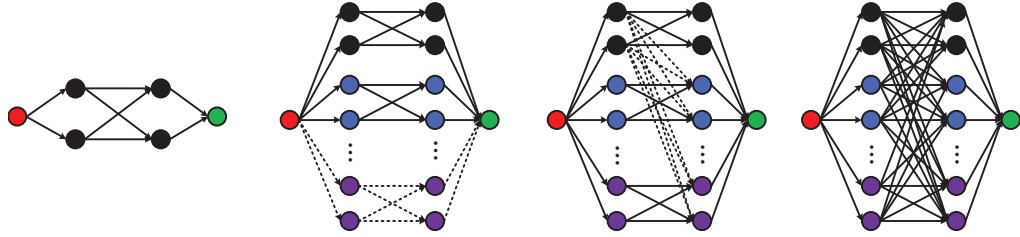

Figure 7: The way to use the proposed RBNN model to train a fully-connected DNN classifier

Figure 8 shows the scaling of test errors over the recursive iterations. The total data storage constraint and structure of the initial neural network in this experiment are the same as ones of the experiment in Sec. 5.1, which are 198.8kB and $784\times200\times200\times10$, respectively. The first three iterations expands the DNN in the tiled manner and the last four iterations adds weights that connect the tiles. In the forth iteration, the neurons of the first hidden layer of the first tile are connected to the neurons in the second hidden layers of all the other tiles, making the DNN 1/4-connected. The bit-width of weights are 7 bits. This is because fewer weights are added than the first three iterations. In following iterations, the hidden layers of the first hidden layer of the rest of the tiles are connected to the hidden neurons of the second hidden layer in the same way as the forth iterations. Figure 8 shows that the fully-connected DNN classifier has the similar accuracy performance as the one has tiled structure in Sec. 5.1.

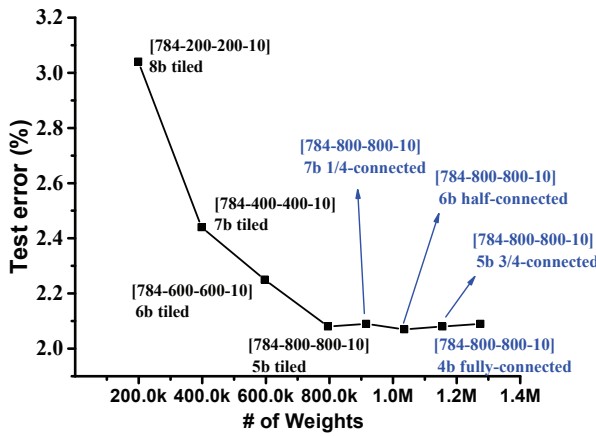

Figure 8: Performance of fully-connected DNN generated by the proposed RBNN model

.

