# OpenReview forum: "Recursive Binary Neural Network Learning Model  with 2-bit/weight Storage Requirement"
_ICLR.cc/2018/Conference — Reject_

### Official Review · AnonReviewer1 · 2017-11-26
**This work suggest how to train a NN in incremental way so for the same performance less memory is needed or for the same memory higher performance can be achieved.**

**Rating:** 7
**Confidence:** 4

**Review:**

The idea of this work is fairly simple. Two main problems exist in end devices for deep learning: power and memory. There have been a series of works showing how to discretisize neural networks. This work, discretisize a NN incrementally. It does so in the following way: First, we train the network with the memory we have. Once we train and achieve a network with best performance under this constraint, we take the sign of each weight (and leave them intact), and use the remaining n-1 bits of each weight in order to add some new connections to the network. Now, we do not change the sign weights, only the new n-1 bits. We continue with this process (recursively) until we don't get any improvement in performance.

Based on experiments done by the authors, on MNIST, having this procedure gives the same performance with 3-4 times less memory or increase in performance of 1% for the same memory as regular network.

I like the idea, and I think it is indeed a good idea for IoT and end devices. The main problem with this method that there is undiscussed payment with current hardware architectures. I think there is a problem with optimizing the memory after each stage was trained. Also, current architectures do not support a single bit manipulations, but is much more efficient on large bits registers. So, in theory this might be a good idea, but I think this idea is not out-of-the-box method for implementation.

Also, as the authors say, more experiments are needed in order to understand the regime in which this method is efficient. To summarize, I like this idea, but more experiments are needed in order to understand this method merits.

---

> ### Author Response · Authors · 2018-01-03
> **Discussion on computation payment, hardware implementation and more experiments are added**
>
> Thank you for your insightful reviews helping us improve our paper. More analysis and discussion about computation payment and hardware implementation are added in the revised paper. And we answer your questions as follows.
>
> The computation payment of the proposed RBNN is very small compared to the energy saving it brings. First of all, from the results on arithmetic complexity Table 2, it is noticed that the extra computation brought by the proposed RBNN are shift and add, and multiplication computation are the same for RBNN and conventional BNN. Since multiplication has much more overhead than add and shift, the final computation increase is not significant. Secondly, it has been proved that for fully connected NN systems, data access costs the majority of the energy overhead. The proposed RNN model reduce the data storage requirement so the system only need to fetch data from on-chip SRAM during training. According to the quantitative analysis added in Table2 and Section 5.3, this saves around 100x energy compared to conventional BNN which has to fetch weights from off-chip DRAM.
>
> The single-bit manipulation can be implemented by very simple hardware logic. We added Appendix A to the revised paper to illustrate the implementation of bit-wise operation of weights. The main idea is fetching complete weights from weight storage. And use mask code to separate fixed bits and plastic bits. After plastic bits are updated, they are concatenated to fixed bits through XOR operation and write back to data storage. This implementation only requires simple AND and XOR operation at the very beginning and end of each training epoch, so the extra energy consumption is very small.
>
> The results of applying the proposed RBNN model to CNNs on MNIST benchmark and to MLP-like DNNs on AURORA 4 benchmark are added in Appendix B and C, respectively. We really appreciate your suggestions to validate the proposed RBNN model more.

---

### Official Review · AnonReviewer2 · 2017-11-27
**Not ready yet; needs more work**

**Rating:** 6
**Confidence:** 3

**Review:**

There could be an interesting idea here, but the limitations and applicability of the proposed approach are not clear yet. More analysis should be done to clarify its potential. Besides, the paper seriously needs to be reworked. The text in general, but also the notation, should be improved.

In my opinion, the authors should explain how to apply their algorithm to more general network architectures, and test it, in particular to convnets. An experiment on a modern dataset beyond MNIST would also be a welcome addition.

Some comments:
- The method is present as a fully-connected network training procedure. But the resulting network is not really fully-connected, but modular. This is clear in Fig. 1 and in the explanation in Sect. 3.1. The newly added hidden neurons at every iteration do not project to the previous pool of hidden neurons. It should be stressed that the networks end up with this non-conventional “tiled” architecture. Are there studies where the capacity of such networks is investigated, when all the weights are trained concurrently.

- It wasn’t clear to me whether the memory reallocation could be easily implemented in hardware. A few references or remarks on this issue would be welcome.

- The work “Efficient supervised learning in networks with binary synapses” by Baldassi et al. (PNAS 2007) should be cited. Although usually ignored by the deep learning community, it actually was a pioneering study on the use of low resolution weights during inference while allowing for auxiliary variables during learning.

- Coming back my main point above, I didn’t really get the discussion on Sect. 5.3. Why didn’t the authors test their algorithm on a convnet? Are there any obstacles in doing so? It seems quite important to understand this point, as the paper appeals to technical applications and convolution seems hard to sidestep currently.

- Fig. 3: xx-axis: define storage efficiency and storage requirement.

- Fig. 4: What’s an RSBL? Acronyms should be defined.

- Overall, language and notation should really be refined. I had a hard time reading Algorithm 1, as the notation is not even defined anywhere. And this problem extends throughout the paper.
For example, just looking at Sect. 4.1, “training and testing data x is normalized…”, if x is not properly defined, it’s best to omit it;  “… 2-dimentonal…”, at least major typos should be scanned and corrected.

---

> ### Author Response · Authors · 2018-01-03
> **Hardware implementation and application on CNNs and other benchmarks are added**
>
> Thank you very much for your insightful comments. We really appreciate your comments which help us to improve the draft. We fixed typos, revised the paper so as to reduce confusion, and also add relevant references including those suggested by the reviewer. Below are our answers to your other questions.
>
> About the topology of the generated neural network, firstly, we correct our presentation of "fully-connected" based on the fact that our RBNN trained the fully-connected structure for the 1-hidden layer case and the tiled structure for the 2-hidden layer case. However, we'd like to point out that in all the experiments in the paper, the results of conventional BNNs that are compared to ones of the proposed model are all fully-connected. We tested the RBNN to train the fully-connected structure for the 2-hidden layer case, but we do not see much difference in terms of the accuracy and storage-requirement trade-off. Still we added this results to Fig. 8.
>
> For hardware implementation, we added Appendix A to illustrate the hardware implementation of memory reallocation.  It describes multi-weight operations where each weight takes one to k bits during the training process based on the RBNN model. The main idea is to fetch multiple weights packed in one 8-bit word from data storage (SRAM) and to use a mask to separate already-trained weights (bits) and plastic weights (bits). After finishing training, we use XOR operation to pack once-plastic bits and the fixed bits into a word and store it in the data storage. This mapping requires bit-wise AND and XOR operations, which are supported in CPU,GPU, custom circuits and also FPGAs, at the very beginning and the end of each training epoch. Therefore, the extra energy consumption is minimal. It also allows us to use the existing SRAM macros without modification. We added this discussion to Sec 5. 3 in the revised paper.
>
> The CNN with the proposed RBNN model is tested and the results are shown the revised paper (Appendix B and C), We added an experimental result on the application of our RBNN on the LeNet CNN performing MNIST benchmark. We also added the experimental result on the application of our RBNN on the MLP-like DNN performing voice activity detection benchmark (AURORA 4). These new results confirm that our RBNN can improve the trade-off between weight storage requirement and accuracy trade-off by the similar amount as the original results from the MLP and the MINST test case.

---

### Official Review · AnonReviewer3 · 2017-11-28
**Nice trick on reusing non-sign bits to recursively add more weights during training, but high computation cost and ideally need more experiments**

**Rating:** 5
**Confidence:** 3

**Review:**

Summary: The paper addresses the issue of training feed-forward neural networks with memory constraints. The idea is to start by training a very small network, binarise this network, then reuse the non-signed bits of the binarised weights to add/store new weights, and recursively repeat these steps. The cost of reducing the memory storage is the extra computation. An experiment on MNIST shows the efficacy of the proposed recursive scheme.

Quality and significance: The proposed method is a combination of the binarised neural network (BNN) architecture of Courbariaux et al. (2015; 2016) with a network growing scheme to reduce the number of bits per weight. However, the computation complexity is significantly larger. The pitch of the paper is to reduce the "high overhead of data access" when training NNs on small devices and indeed this seems to be the case as shown in the experiment. However, if the computation is that large compared to the standard BNNs, I wonder if training is viable on small devices after all. Perhaps all aspects (training cost [computation + time], accuracy and storage) should be plotted together to see what methods form the frontier. This is probably out of scope for ICLR but to really test these methods, they should be trained/stored on a real small device and trained/fine-tuned using user data to see what would work best.

The experiment is also limited to MNIST and fully connected neural networks.

---

> ### Author Response · Authors · 2018-01-03
> **The extra computation complexity can be easily offset by the notably less amount of off-chip data storage access. And the proposed model can achieve ~100X energy saving for training. New experiments on CNN are also added.**
>
> Thank you for your insightful comments. Your concern on the extra computation overhead of our proposed model is valid. However, we'd like to point out that it is not as significant compared to our benefit in data access. In terms of energy, compared to conventional BNN, the proposed model needs notably less amount of off-chip data storage access, which easily offsets the extra computation cost with a large margin. To elaborate this issue more, we added quantitive analysis in Sec. 5.3, Table 2 and Table3 in the revised paper.
>
> First, in a BNN, the main bottleneck is data access overhead rather than computation. This is because the use of binary information of weights reduces computational complexity. The proposed model reduces the data storage size so that it can store all the weights in the on-chip SRAM. This reduces energy consumption significantly because accessing data from off-chip DRAM and FLASH consumes at least 2 orders of magnitude more energy than SRAM. Conventional BNN systems have to store and fetch data from off-chip DRAM and FLASH. Our quantitative energy analysis, added in Sec. 5.3, shows the proposed RBNN can save at least 100X training energy compared to conventional BNN.
>
> Second, the proposed model only increases the number of add and shift operations roughly two times for the neural networks having the same number of hidden units (Table 2), whereas it does not increase the number of multi-bit multiplications as compared to conventional BNNs. Note that this multi-bit multiplication is used to calculate gradients. In both RBNN and BNN, the multiplications between inputs/activations and weights are replaced with sign change operations. Multiplication is much more costly than add and shift operations. Thus, it is important not to increase the number of multiplications.
>
> And the evaluation of the proposed model on CNN classifying MNIST benchmark and DNN classifying AURORA 4 VAD benchmark is added in the Appendix B and C in the revised paper, respectively

---

### Decision · Program_Chairs · 2018-01-29
**ICLR 2018 Conference Acceptance Decision**

**Decision:**

Reject

**Comment:**

This is an interesting paper and addresses an important problem of neural networks with memory constrains. New experiments have been added that add to the paper, but the full impact of the paper is not yet realised, needing further exploration of models of current practice, wider set of experiments and analysis, and additional clarifying discussion.